# POSITIVE TRANSFER OF PRIOR KNOWLEDGE IN DEEP REINFORCEMENT LEARNING VIA REWARD SHAPING

## ABSTRACT

Effective learners improve task performance and acquire new skills more efficiently by leveraging related prior knowledge. Reward shaping is central to many such approaches and facilitates knowledge transfer. However, misidentifying or misusing prior knowledge can impair learning. To tackle this challenge, we propose a novel shaping method, Target value As Potential (TAP), which uses critic target value as the potential to operate within the canonical Potential-Based Reward Shaping (PBRS) framework. It integrates readily with policy-gradient deep reinforcement learning algorithms and requires only minor modifications to existing training pipelines. This endows TAP with the unique combination of policy invariance and simplicity in implementation, distinguishing it from many model-based methods. Our qualitative analysis and empirical evaluations demonstrate that TAP accelerates convergence compared to baseline DRL algorithms. Moreover, empirical results show that TAP leads to higher cumulative returns. We evaluate TAP-augmented TD3 and D4PG across a range of tasks in the DeepMind Control Suite. TAP significantly improves performance over the original TD3 and D4PG and consistently outperforms other reward shaping methods, including Heuristic-Guided Reinforcement Learning (HuRL) and Dynamic Potential-Based Reward Shaping (DPBRS).

## 1 INTRODUCTION

Reward shaping is a widely used technique that accelerates learning by injecting prior knowledge. Within the standard MDP framework, potential-based reward shaping (PBRS) is the **unique** framework ensuring policy invariance: an additive shaping term preserves the set of optimal policies if and only if it is a potential difference Ng et al. (1999). While this classical shaping mechanism is well understood and holds great promise for improving learning performance, the central **challenge** lies in identifying and encoding effective, transferable prior knowledge into a shaping reward.

To develop useful priors for shaping, most existing efforts have focused on two major families of priors. 1) **external priors**. Shaping signals can be engineered from external structure—e.g., solving simplified or relaxed task variants or using planning heuristics Hoffmann & Nebel (2001); Buffet & Hoffmann (2010); Adamczyk et al. (2023)—or imported via policy transfer from previously learned tasks Brys et al. (2015). These approaches can be effective under the assumption that the priors are beneficial to learning, but performance depends critically on the quality and relevance of the prior. As shown in (Koenig,1996) Koenig & Simmons (1996) that the choice of prior knowledge representation whether through rewards or value initialization can profoundly impact learning outcomes, both positively and negatively. 2) **Data- or experience-driven priors**. Alternatively, the shaping signal can be learned from data—by fitting auxiliary reward models or potentials Brys et al. (2015); Koenig & Simmons (1996); Burda et al. (2018); Devidze et al. (2022); Ma et al. (2024); Hu et al. (2020); Grzes & Kudenko (2009) —or derived from intrinsic-motivation bonuses that promote novelty and state-space coverage (e.g., RND, count-based exploration, exploration studies) Burda et al. (2018); Tang et al. (2017); Mavor-Parker et al. (2022). While these signals often improve exploration, in stochastic or noisy environments they can overvalue distractors and pull agents toward irrelevant high-novelty regions, delaying convergence or yielding suboptimal behavior. Canonical curiosity methods such as ICM and VIME exhibit similar trade-offs when mis-scaled or mis-specified Pathak et al. (2017); Houthooft et al. (2016).

Common in most of the above methods, they require learning a separate reward model Ma et al. (2024); Hu et al. (2020); Grzes & Kudenko (2008) or a separate potential function model of the environment Grzes & Kudenko (2009); Ng et al. (1999) , which can introduce additional complexity, computational overhead, and potential inaccuracies that may hinder the overall learning performance. Another concern of many of these methods is their lacking of policy invariance guarantee if they are not potential-based Burda et al. (2018); Tang et al. (2017); Badnava et al. (2023). The key challenge, therefore, is to reduce the heavy reliance on prior domain knowledge or human feedback, while still ensuring data efficiency and preserving policy invariance as in PBRS.

**Contributions of this work.** We introduce Target value As Potential (TAP), a novel reward-shaping method within the PBRS framework that leverages the target $Q$-network in off-policy DRL. Unlike many task-specific approaches that rely on reward models, TAP offers several advantages: (1) it is simple and easy to be integrated seamlessly into standard policy-gradient pipelines without adding hyperparameters; (2) it accelerates learning, as evidenced by both qualitative analyses and empirical results; (3) it promotes exploration and yields higher returns; and (4) it is implementation-flexible, enhancing performance in both same-task and cross-task transfer settings.

## 2 RELATED WORK

**Potential Based Reward shaping (PBRS)** is a special form of reward shaping with the desirable property of ensuring policy invariance with the use of shaped reward Ng et al. (1999); Brys et al. (2015). Few existing shaping schemes provide such guarantee. To account for prior knowledge, it involves modifying an original reward function by adding a shaping reward which is expressed as a difference between the potential values. Specifically, the shaping reward $F(s_k, s_{k+1}) = \gamma \Phi(s_{k+1}) - \Phi(s_k)$, where $\Phi$ is a potential function reflecting prior knowledge that can be used to provide insights on agent-environment interacting dynamics. The potential function often required strong domain knowledge and obtained by solving a relaxed version of the original problem Hoffmann & Nebel (2001); Richter & Westphal (2010); Hu et al. (2020). A dynamic extension of PBRS (DPBRS) was later introduced in Badnava et al. (2023), where the potential function is derived from episode-level performance. Potential-based reward shaping for intrinsic motivation (PBIM) Forbes et al. (2024) uses a similar idea as DPBRS by accumulating the reward signal into a state-based potential. While these approaches show promise, however, they usually compromise the policy invariance guarantee. Alternatively, Heuristic-guided RL (HuRL) Cheng et al. (2021) may use expert demonstrations, exploratory datasets, and engineered guidance to construct heuristics that represent an initial estimate of the long-term return of states. This approach aims at effectively using heuristics to transform the problem into a shorter-horizon subproblem using a mixing coefficient to mimic the original task. The effectiveness of HuRL, however, is contingent upon the quality of the available heuristics, which may adversely impact learning when using a poorly constructed heuristic. Overall, the potential-based reward shaping (PBRS) framework is highly promising. Nevertheless, practical methods for applying it to complex, high-dimensional continuous-control tasks—without training an auxiliary reward model—remain limited. Identifying an effective potential function within the PBRS framework also remains challenging.

**Model-Based Reward Shaping**. **How to** obtain trust worthy and efficient prior knowledge has always been at the center of recent PBRS research. Existing approaches to representing prior knowledge is often derived from data Grzes & Kudenko (2008); Harutyunyan et al. (2015) by learning value functions such as BARFI Gupta et al. (2023) which utilizes a bilevel optimization objective to learn behavior-aligned reward functions. These functions incorporate auxiliary rewards derived from designer heuristics and domain knowledge alongside primary environmental rewards. The Self-Tuning Networks Stadie et al. (2020), on the other hand, adapt the intrinsic reward function parameters to guide the policy towards more effective learning. ROSA Mguni et al. (2023) aims at automating reward shaping espiecially for addressing reward conditions that are sparse or uninformative. The problem is solved from a two-player zero-sum Markov game. In contrast, ReLara Ma et al. (2024) is set up in as a coopeartive game to self-learn reward models based on state transitions without relying on domain knowledge or human intervention. However, these learned reward models may introduce additional estimation errors and remain **task-specific**. More importantly, they often cannot be effectively transferred across similar tasks, as the learned reward function is tightly coupled to the original task's state distribution, dynamics, and feature representation, and they also lack a theoretical guarantee of policy invariance.

**Exploration in Reward Shaping**. Reward shaping has also been used to encourage exploration by incorporating exploration bonuses—additional rewards designed to capture state novelty and promote comprehensive coverage of the environment Devidze et al. (2022); Sun et al. (2022); Tang et al. (2017). Two broad approaches are common. Exploration-driven reward schemes can overpower the task reward, causing agents to linger in unproductive, distracting regions of the state space. Curiosity-driven methods mitigate this by granting intrinsic rewards for prediction error or surprise, encouraging visits to novel or unpredictable states while better balancing the task objective. For instance, Pathak et al. (2017) generated intrinsic rewards based on prediction errors of the agent's own actions within a learned feature space. Additionally, Mezghani et al. (2023) utilized pre-collected data with hindsight relabeling to better understand environmental structure and dynamics. Nevertheless, these curiosity-driven methods still face challenges such as sensitivity to noise, the risk of overfitting to irrelevant state features, and potential inefficiency in environments with inherently complex dynamics. Additionally, these methods are not readily shown to be policy invariant under knowledge transfer.

**Knowledge Transfer.** Contextual policy transfer Gimelfarb et al. (2021) introduces a Bayesian mixture-of-experts model to learn state-dependent posterior distributions over source task dynamics. It has improved sample efficiency and performance across various benchmark tasks. However, it faces challenges in complex environments when relevant source policies are not readily available and an accurate estimation of dynamics is not guaranteed. The Policy Teaching Framework (PTF) Yang et al. (2020) models multi-policy transfer as an option learning problem to dynamically select and terminate the use of source policies based on their performance to effectively accelerate the learning process and improves final performance. However, PTF assumes that source policies are at least partially useful, yet, it lacks a formal measure to quantify policy relevance. The Single Episode Policy transfer Yang et al. (2019) is based on a rapid estimation of underlying latent variables of test dynamics using a small fraction of the test episode. However, it assumes early detectability of dynamic differences, relies on well-designed probe policies and simulators which require strong domain knowledge. The Successor Features method Barreto et al. (2017a) centers on value function representation by decoupling the dynamics of the environment from the rewards. Its generalized policy improvement utilizes a set of source policies to provide performance improvement. However, the approach relies on an assumption that the environment dynamics preserves across tasks. Additionally, the reliance on predefined or learned feature representations can pose challenges in environments with complex or high-dimensional state spaces. MAXQINIT Abel et al. (2018) is a value-function-based transfer method that aims to reduce learning expenditures in new tasks while preserving Probably Approximately Correct guarantees. However, successful demonstrations of sample efficiency was limited to relatively simple lifelong RL tasks.

## 3 METHOD

**Background.** We consider a reinforcement learning agent interacts with its environment in discrete time. At each time step $k$, the agent observes a state $s_k \in \mathbf{S}$ and select an action $a_k \in \mathbf{A}$ based on its policy $\pi : \mathbf{S} \to \mathbf{A}$, namely, $a_k = \pi(s_k)$, and receives a scalar reward $r(s_k, a_k) \in \mathbf{R}$ (use $r_k$ as short hand notation).

Evaluation of a policy $\pi$ is performed using the expected return after taking an action $a_k$ in state $s_k$ following the policy $\pi$:

$$Q^\pi(s_k, a_k) = \mathbb{E}[R_k | s_k, a_k]$$

$$\text{where } R_k = \sum_{t=k}^{\infty} \gamma^{t-k} r_t,$$

$$s_k \sim p\left(\cdot \mid s_{k-1}, a_{k-1}\right),$$

$$a_k = \pi\left(s_k\right), \tag{1}$$

with $0 < \gamma < 1$. For an actor-critic method (Lillicrap et al. (2015); Fujimoto et al. (2018); Haarnoja et al. (2018)), the policy ($\pi_\phi$) is represented by a policy network (the actor) with parameters $\phi$, and the state-action value function $Q_\theta$ is represented by a critic network (the critic) with parameters $\theta$. Consequently, the respective target networks for the actor and the critic are represented by $\pi_{\phi'}$ and $Q_{\theta'}$.

Most actor-critic methods are based on temporal difference (TD) learning (Sutton & Barto (2018a); Si et al. (2004)) that updates $Q$ estimates by minimizing the TD error, which is the difference between a target value and an estimated critic value where the target value $y_k$ is:

$$y_k = r_k + \gamma Q_{\theta'}(s_{k+1}, \pi_{\phi'}(s_{k+1})). \tag{2}$$

Thus the critic value $Q_\theta$ is updated by minimizing the loss function $(L(\theta))$ with respect to the critic weights $\theta$:

$$L(\theta) = \mathbb{E}_{s_k \sim p_\pi, a_k \sim \pi}[(y_k - Q_\theta(s_k, a_k))^2]. \tag{3}$$

where $s_k \sim p_\pi$ is the state probability induced by the policy $\pi$. The actor weights ($\phi$) can be updated by deterministic policy gradient algorithms with the gradient described as $\nabla_\phi J(\phi)$ below (Silver et al. (2014)).

$$\nabla_\phi J(\phi) = \mathbb{E}_{s \sim p_{\pi_\phi}} \left[ \nabla_a Q_\theta(s, a)|_{a=\pi_\phi(s)} \nabla_\phi \pi_\phi(s) \right]. \tag{4}$$

**The TAP method.** We propose to use the target critic network parameterized by $\theta'$ as the potential (TAP) function. Let the target critic be $Q_{\theta'}$ and the target actor (policy) be $\pi_{\phi'}$. We formulate the potential function as

$$\Phi(s_k) = Q_{\theta'}(s_k, \pi_{\phi'}(s_k)). \tag{5}$$

Accordingly, our shaping reward signal $\bar{r}$ as in potential-based reward shaping becomes

$$\bar{r}_k = \gamma Q_{\theta'}(s_{k+1}, \pi_{\phi'}(s_{k+1})) - Q_{\theta'}(s_k, \pi_{\phi'}(s_k))). \tag{6}$$

The TAP method can be used in different learning scenarios. The two major transfer applications can be as follows. 1) Using the potential function $\Phi(s)$ in Equation (5) to transfer target network value during **same-task** learning can improve **same-task** learning performance. 2) The potential function $\Phi(s)$ in Equation (5) can also be from externally learned target network value to improve **cross task** learning. Additionally, TAP can be flexibly implemented to account for prior knowledge.

In all the above use cases of TAP, the shaped reward $r'_k$ is

$$r'_k = r_k + \bar{r}_k. \tag{7}$$

Then the knowledge-shaped value function $\mathbb{Q}^\pi(s_k, a_k)$ for all state-action pairs and for policy $\pi$ is

$$\mathbb{Q}^\pi(s_k, a_k) = \mathbb{E}[r'_k + \sum_{j=1}^{\infty} \gamma^j r'_{k+j}]. \tag{8}$$

From Equations (8) and (6) we have the following relationship between shaped $Q$ value and the $Q$ value without shaping as:

$$\mathbb{Q}^\pi(s_k, \pi(s_k)) + Q_{\theta'}(s_k, \pi_{\phi'}(s_k))) = Q^\pi(s_k, \pi(s_k)). \tag{9}$$

Henceforth, the knowledge-shaped value function can be updated following the general policy gradient procedure as described in Equations (2) and (3).

## 4 THEORETICAL ANALYSIS

Theorem 2 below provides a qualitative analysis to show that TAP reward shaping yields faster learning than training without shaping. To contextualize this result, we first present Theorem 1—an adaptation within well-established framework—primarily to set the stage for Theorem 2. We also include further insights into TAP, and we verify that TAP preserves policy invariance; full details and all proofs are provided in Appendix E.

**Theorem 1** (Convergence and optimality of TAP). Let $\{\pi_i\}_{i \geq 0} \subseteq \Pi$ be the sequence of policies obtained from repeated application of TAP policy evaluation and TAP policy improvement. Then $\{\pi_i\}_{i \geq 0}$ converges to an optimal policy $\pi^*$. Moreover, for every $(s, a) \in \mathbb{S} \times \mathbb{A}$, and every $\pi \in \Pi$, $\mathbb{Q}^{\pi^*}(s, a) \geq \mathbb{Q}^\pi(s, a)$. Consequently, $\mathbb{Q}^{\pi^*} = \mathbb{Q}^*$, the optimal value function.

*Proof.* Details are provided in Appendix E. □

Next, we examine how TAP may benefit learning. For that purpose, we consider optimal policy $\pi^*(s)$ with optimal value $Q^{\pi^*}(s, \pi^*(s)) = Q^*(s, \pi^*(s))$, which denotes the optimal $Q$ value without using reward shaping, and which satisfies the Bellman optimality equation:

$$Q^{\pi^*}(s_k, \pi^*(s_k)) = \mathbb{E}[r_k + \gamma Q^{\pi^*}(s_{k+1}, \pi^*(s_{k+1}))]. \tag{10}$$

**Theorem 2.** Let the stage reward $r_k$ of the target task be bounded by $r_{max}$. Let $\mathbb{Q}(s, a)$ and $Q(s, a)$ (with respective short hand notation $\mathbb{Q}$ and $Q$, and similarly thereafter) as the $Q$-value functions with and without reward shaping, respectively. Assume that the potential function $\Phi = Q_{\theta'}$ containing prior knowledge remains constant, and also that $0 < Q_{\theta'} \le Q^*$. Set $Q_0 = \mathbb{Q}_0 = 0$. Let $q$ be the $Q$-value that can be reached at step $n_s$ with shaping, and at step $n_{ns}$ without shaping, respectively. Let $\epsilon = \|q - Q^*\|$, we have the following results.

1. $n_{ns} \le \ln\left(\dfrac{\epsilon}{\|Q^*\|}\right) / \ln(\gamma)$.

2. $n_s \le \ln\left(\dfrac{\epsilon}{\|Q_{\theta'} - Q^*\|}\right) / \ln(\gamma)$.

3. Let $\bar{n}_{ns}$ and $\bar{n}_s$ be the upper bounds of $n_{ns}$ and $n_s$, respectively. Then $\bar{n}_{ns} > \bar{n}_s$.

*Proof.* Details are provided in Appendix E. □

**Remark 1.** 1) Theorem 2 suggests that it take less time to reach the same reward level for learning with our reward shaping method TAP than without shaping. 2) Now consider a realistic situation where the priors are useful but not perfect. We provide some insight from the perspective of value decomposition. We decompose the $Q^*$ around $Q_{\theta'}$ as follows,

$$Q^* = \underbrace{Q_{\theta'}}_{\text{(a useful prior)}} + \underbrace{\left(Q^* - Q_{\theta'}\right)}_{\text{(Difference to be learned)}},$$

where $\mathbb{Q}^* = Q^* - Q_{\theta'}$ from Equation (9). The agent's learning may be viewed as fine-tuning (to learn the difference term) rather than learning from scratch. As such, the better the quality of the prior, the less to learn.

**Remark 2 (Insights from a simple maze problem).** To shed some light on Theorem 2, as well as Remark 1, we provide some empirical results using a simple maze problem. Further details of the environment set up are provided in Appendix B. Using this maze problem, we show how the quality of priors affect learning performance. To simulate that, for example, a poor quality prior $Q_{\theta'}$ may be obtained at 10k steps of learning while a high quality prior $Q_{\theta'}$ may be obtained at 250k steps of using baseline $Q$-learning. We use this example to answer the following questions:
**Q**1. How the quality of priors affect learning speed and cumulative reward?
**Q**2. How the quality of priors affects exploration during learning?

**Q**1. **Higher valued priors are associated with accelerated learning and higher rewards**. This evaluation is based on the original $Q$-learning to provide insights by comparisons between $Q$-learning with and without TAP. Specifically, $Q$ tables are obtained at different learning stages (equally spaced at 50k steps), which are used in TAP to simulate varying quality of the priors (the more stages involved the higher prior quality).

Figure 1b shows the effect of the quality of the priors on transfer learning performance. Clearly, a prior obtained at a later stage is associated with faster convergence to the optimal value. At time step 250k, we see instant convergence with success rate of 1. This result corroborates Theorem 2 that using high quality priors reduces learning expenditures with reduced learning time. Figure 1 results align with Remark 1 above. We can clearly see this effect again in the DRL benchmark results in Q3 and Figure 2.

**Q**2. **TAP improves exploration of high-value regions.** Further analysis of Figure 1c, d, e reveals that utilizing a $Q$ table with prolonged learning reduces the likelihood of agents entering the trap

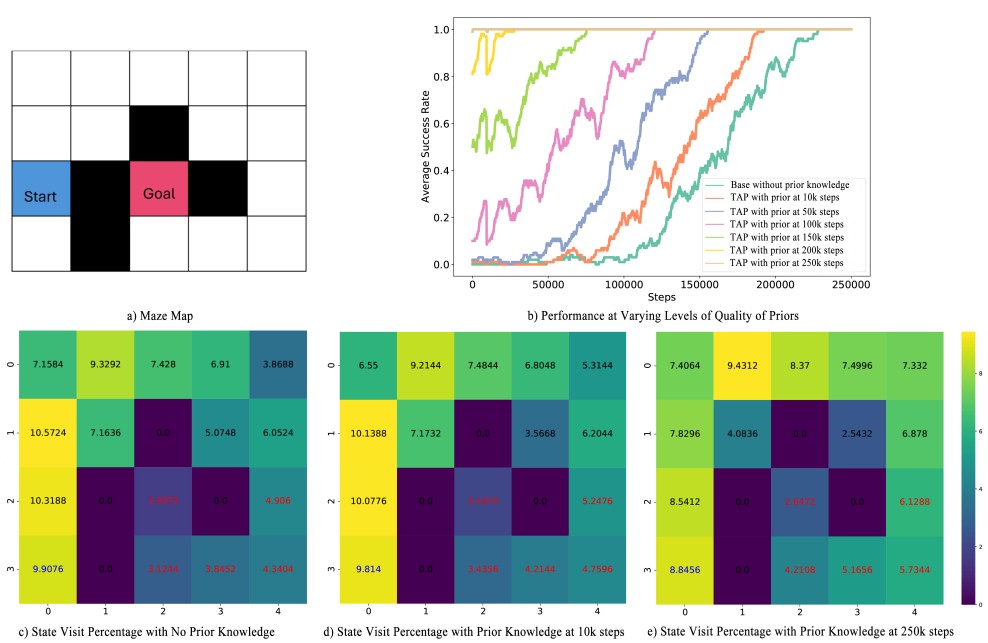

Figure 1: With TAP for reward shaping, the learning performance improves in proportion to the quality of the priors (the learned $Q$-values) obtained at different stages of learning the maze-solving task. a) The maze, where the blue block: initial state; the red block: terminal state; the black blocks: walls. Reaching terminal state receives a reward 10 or $r_k = 10$, otherwise $r_k = -1$. b) Learning success rate (averaged over 10 evaluation trials) of $Q$-learning with TAP. The $Q$-tables at every 50k steps are incorporated as prior knowledge for TAP. c) Heat map of state visits for base $Q$-learning without prior knowledge. The numbers in the blocks represent the percentage of visit times over total steps. d) Same as case c) except using TAP with a learned $Q$ table from the base method up to 10k steps. e) Same as d) except the learned $Q$ table is up to 250k steps.

state at the bottom left corner $(0, 3)$. Conversely, agents become more likely to visit states with higher values such as $(2, 2)$, $(2, 3)$, $(3, 3)$, $(4, 3)$, and $(4, 2)$. This observation indicates that TAP has enhanced exploration in high-value regions while reduced exploration in low-value areas such as trap states. The advantage of TAP becomes more pronounced as the prior knowledge approaches the optimal value.

## 5 EXPERIMENTS AND RESULTS

We evaluate our TAP on 20 different tasks in 8 benchmark environments in DMC, including cartpole, quadruped, fish, hopper, finger, walker, dog, humanoid. Our evaluations include both **same-task** transfer as well as **cross-task** transfer from easier to more difficult tasks. Details of the implementation, training, evaluation procedures, and code are provided in Appendix B. We use two high-performing DRL algorithms, TD3 and D4PG, as baseline methods to be augmented with TAP. We compare TAP performance with two benchmark reward shaping methods, HuRL Cheng et al. (2021) and DPBRS Badnava et al. (2023). The former is a principled method for injecting prior knowledge into RL, while the latter uses knowledge extracted from its own learning process to formulate shaping reward signal.

In the following, we present systematic results of using TD3 while D4PG results, which corroborate the findings from TD3, are given in Appendix D due to space limitation. In reporting evaluation results below, we use the following short-form descriptions.

1) **Base**: the original DRL algorithms (TD3, D4PG).

2) **TAP-Self**: TAP using internally generated priors from target network value in Equation (6) during learning of the **same task**.

3) **TAP-Same**: TAP using priors from target network values of different runs in Equation (6) to learn the **same task**.

4) **TAP-Cross**: TAP using externally generated priors in Equation (6) from a simpler task (different from the target task), namely, **cross-task** transfer.

5) **HuRL**: HuRL-MC is used as it is considered the best performing variant Cheng et al. (2021).

6) **DPBRS**: Dynamic Potential-Based Reward Shaping method Badnava et al. (2023).

Our evaluations of TAP aim to quantitatively address the following questions:

**Q**3. How effectively TAP improves Base DRL methods for learning the same task?
**Q**4. How effectively TAP improves Base DRL methods for cross-task learning?
**Q**5. How does TAP encourage exploration?

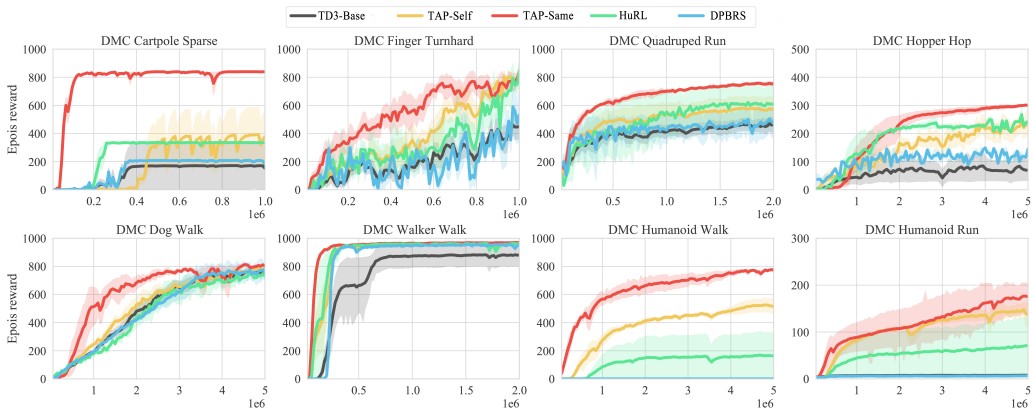

Figure 2: Systematic evaluation of TAP with TD3 as Base for **same-task** transfer in 7 DMC environments. The shaded regions represent the 95 % confidence range of evaluations over 10 seeds. The $x$-axis is the number of steps. The full set of results evaluated for 20 benchmark tasks in all 8 environments are in Appendix C. Corresponding results of D4PG are in Appendix D.

## 5.1 MAIN RESULTS

**Q**3. **TAP boosts Base method performance in learning the same task.** TAP incorporates prior knowledge via two transfer strategies: TAP-Self and TAP-Same. Figure 2 presents partial results of TAP in DMC environments. Full benchmark results are provided in Appendix C, and the corresponding full D4PG results are in Appendix D. The results show that TAP-Self (yellow lines) consistently enhances learning speed, increases total reward, and reduces learning variance compared to its Base method (black lines). Notably, TAP-Same (red lines) further improves performance significantly. TAP is especially promising in complex tasks such as Hopper Hop and Humanoid Walk, where the Base method fails to learn and the HuRL method achieves 20% lower converged rewards.

When compared to the better of the two evaluated reward shaping methods, namely HuRL, our TAP-Same significantly outperforms HuRL across all benchmark tasks. This advantage may be attributed to the factor that HuRL relies on knowledge derived from a prior dataset. In sparse or complex environments, behavior cloning is often ineffective due to inconsistencies in action selection across different policies. Moreover, training a heuristic using basic Monte Carlo regression becomes particularly challenging in such settings Cheng et al. (2021).

When compared to the DPBRS method that uses internally generated prior knowledge, TAP-Self also outperforms DPBRS significantly across all benchmark tasks in all evaluated environments. This improvement may stem from two key reasons: 1) DPBRS heavily relies on current exploration strategy, with the shaped reward basing solely on the current reward and heuristic episodic rewards.

However, in problems with limited reward feedback or high-dimensional spaces, exploration lacks sufficient guidance. As a result, it relies on trial-and-error. 2) DPBRS uses aggregated signals in formulating their shaping signal, which may be problematic in continuous state problems, especially those involving complex dynamics. In contrast, TAP-Self utilizes the current target $Q$ network and target policy to formulate the shaping reward. As learning progresses, TAP leverages dynamic feedback through target network updates to guide exploration more effectively while providing detailed information about the expected returns of different states, thereby ensuring more robust and efficient learning.

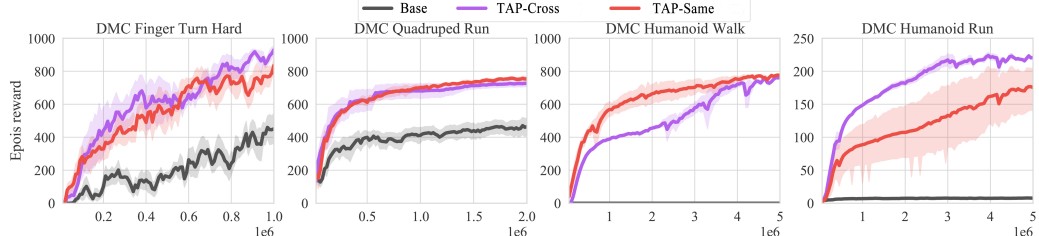

Figure 3: Systematic evaluation of TAP with TD3 as Base in DMC environments for **cross-task** transfer learning. The shaded regions represent the 95 % confidence range of evaluations over 10 seeds. The $x$-axis is the number of steps. Finger Turn hard environment uses transfer from Finger Turn easy; Quadruped Run from Quadruped walk; Humanoid Walk from Humanoid Stand; and Humanoid Run from Humanoid Walk.

**Q**4. **TAP significantly improves performance on challenging tasks by enabling efficient transfer from simpler tasks.** Even though reward shaping has been shown facilitating prior knowledge transfer across different tasks Zhu et al. (2023); Marom & Rosman (2018); Barreto et al. (2017b), those results are only for simple and low dimension tasks such as grid world, pin ball and reacher. The challenges still remain for high dimensional continuous control tasks like humanoid walk and run. We evaluate the following cross-task learning scenarios: 1) Finger Turn Easy to Finger Turn Hard. 2) Quadruped Walk to Quadruped Run. 3) Humanoid Stand to Humanoid Walk. 4) Humanoid Walk to Humanoid Run.

For TD3, existing benchmarks Pardo (2020); Hoffman et al. (2020) report little progress on humanoid tasks; meaningful performance typically requires distributional learning, as in the advanced D4PG of Barth-Maron et al. (2018). In contrast, our TAP methods deliver substantial gains when augmenting baseline TD3: (1) TAP-Self enables successful learning of Humanoid Walk and Run (Figure 2), tasks on which baseline TD3 fails; and (2) building on priors saved from TAP-Self (with TAP-Cross initialized from a simple walking task), TAP-Same and TAP-Cross further improve performance, reaching levels comparable to D4PG, as shown in Figure 3. Notably, TAP achieves the level of performance using a single core, without architectural modifications such as using distributed training (D4PG) with 32 CPU cores (independent actors) as required by the advanced version of Barth-Maron et al. (2018). For the same architecture, a directly comparable result with what we report here can be found from a benchmark report Hoffman et al. (2020). These results demonstrate a capability of TAP that prior reward-shaping methods have struggled to achieve.

In all four cross-task transfer learning experiments, TAP using different simple task knowledge improves the performance of Base methods. Interestingly, for the Finger Turn Hard and Humanoid Run tasks, using external knowledge from respectively simpler tasks (TAP-Cross) proves to be much more beneficial than using external knowledge generated from the respectively **same-task** (TAP-Same). This advantage likely stems from that these tasks (e.g., transitioning from walking to running) share similar underlying low-level dynamics and features—such as balance control, joint coordination, and gait cycle—while primarily differing in high-level features such as movement speed. As a result, transferring external knowledge from simple tasks enables the agent to quickly adapt to more complex behaviors by reusing well-aligned motor primitives.

**Q**5. **TAP encourages exploration in high-value regions while discouraging that in low-value regions.** Figure 4 presents a comparison of state-action-reward data with and without TAP-Self based on 1 million data samples over a simple cartpole environment and a complex humanoid environment. The state and action dimensions are reduced by using T-SNE (available in the sklearn

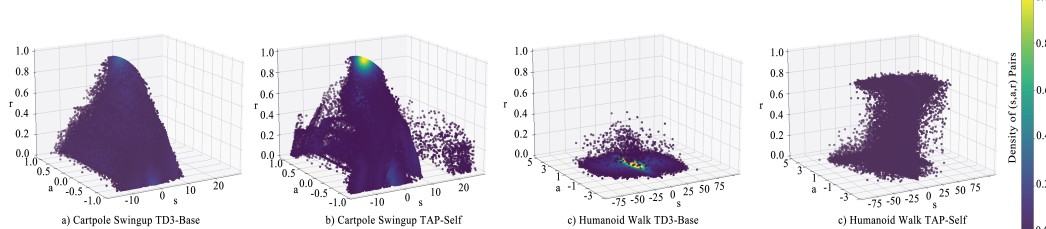

Figure 4: Scatter plot of state-action-reward data to illustrate how TAP enables effective exploration.

Pedregosa et al. (2011) package , a nonlinear dimension reduction technique Van der Maaten & Hinton (2008) to enable visualization. Higher density regions are represented in yellower hues.

In the cart pole environment, TD3 explores primarily within the limited state range of $[-10, 10]$. With TAP, however, it is evident that some low-value regions are avoided, while the range of exploration has expanded into new areas. Furthermore, a focused high-value region is clearly visible. For the more complex humanoid environment, exploration by the Base method has limited to low-value region, while TAP has enabled exploration in much broader areas and into significantly higher reward-value region. The improved exploration by using TAP may be due to that prior knowledge has provided TAP with useful information for policy improvement as only the difference in value in Equation 11 is to be learned. According to Theorem 2, this not only reduces the learning overhead, but also allows learning - when guided by prior knowledge - to be viewed as fine-tuning rather than starting from scratch.

## 6 DISCUSSION AND CONCLUSION

We have introduced a novel TAP-based transfer method that can significantly boost performance of baseline policy gradient methods. It can be easily piggy-backed onto existing policy gradient RL algorithms. The implementation of TAP-based DRL using Equations (6) does not introduce any new hyperparameters. The shaping reward signal can be obtained either externally or internally such that TAP can be used to enhance **same-task** learning or cross-task learning.

**Limitation** Our TAP method demonstrates promising outcomes both theoretically and empirically. But it has two significant limitations. First, the current formulation of TAP assumes that the source and target tasks have the same dimensions in state and action spaces. This is problematic for problems such as transfer learning between quadruped fetch and escape. Second, the TAP-Self usage is limited in the off-policy methods since on-policy method such as PPO does not have the target network to provide the prior.

## 7 ETHICS STATEMENT

All authors, are adhere to the ICLR Code of Ethics.

## 8 REPRODUCIBILITY STATEMENT

Details of network setting, base algorithms, codes are described in Appendix B. Once the paper gets accepted we will release all of them in Github. The theoretical analysis details are in Appendix E.

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

## A  THE USE OF LARGE LANGUAGE MODELS

(i) To search for relevant literature, and (ii) to polish sentences for improved grammar.

## B  IMPLEMENTATION DETAILS

We use PyTorch for all implementations. All results were obtained using our internal server consisting of AMD Ryzen Threadripper 3970X Processor, a desktop with Intel Core i7-9700K processor, and two desktops with Intel Core i9-12900K processor.

**Simple Maze environment Setup:** start state is the initial state and goal is the terminal state. Black cell is the walls. **Reward Setup:** Only receive reward 10 when reach terminal state as $r_k = 10$ if $s = Goal$ else $-1$. **Algorithms Detail:** The Q learning used in this problem is from sutton book Sutton & Barto (2018b). All algorithms and environment codes will be provided after paper get accepted.

**DRL Training Procedure**.

An episode is initialized by resetting the environment, and terminated at max step $T = 1000$. A trial is a complete training process that contains a series of consecutive episodes. Each trial is run for a maximum of $5 \times 10^6$ time steps with evaluations at every $5 \times 10^4$ time steps for complex benchmarks e.g. humanoid, hopper, and dog. For medium complexity locomotion task such as quadruped and fish, each trial is run for a maximum of $2 \times 10^6$ time steps with evaluations at every $2 \times 10^4$ time steps. For simple task such as cartpole and finger, each trial is run for a maximum of $1 \times 10^6$ time steps with evaluations at every $1 \times 10^4$ time steps. Each task is reported over 10 trials where the environment and the network were initialized by 10 random seeds, $(0 - 9)$ in this study.

For each training trial, to remove the dependency on the initial parameters of a policy, we use a purely exploratory policy for the first 8000 time steps (start timesteps). Afterwards, we use an off-policy exploration strategy, adding Gaussian noise $\mathbb{N}(0, 0.1)$ to each action.

**Evaluation Procedure**.

Every $5 \times 10^4$, $2 \times 10^4$ and $1 \times 10^4$ time steps training depends on task complexity, we have an evaluation section and each evaluation reports the average reward over 5 evaluation episodes, with no exploration noise and with fixed policy weights. The random seeds for evaluation are different from those in training which each trial, evaluations were performed using seeds $(seeds + 100)$.

**Network Structure and optimizer**.

**TD3**. The actor-critic networks in TD3 are implemented by feedforward neural networks with three layers of weights. Each layer has 256 hidden nodes with rectified linear units (ReLU) for both the actor and critic. The input layer of actor has the same dimension as observation state. The output layer of the actor has the same dimension as action requirement with a tanh unit. Critic receives both state and action as input to THE first layer and the output layer of critic has 1 linear unit to produce $Q$ value. Network parameters are updated using Adam optimizer with a learning rate of $10^{-3}$ for simple control problems. After each time step $k$, the networks are trained with a mini-batch of a 256 transitions $(s, a, r, s')$.

**D4PG**. Same with the actor-critic networks in D4PG are implemented by feedforward neural networks with three layers of weights. Each layer has 256 hidden nodes with rectified linear units (ReLU) for both the actor and critic. The input layer of actor has the same dimension as observation state. The output layer of the actor has the same dimension as action requirement with a tanh unit. Critic receives both state and action as input to THE first layer and the output layer of critic has a distribution with hyperparameters for the number of atoms $l$, and the bounds on the support $(V_{min}, V_{max})$. Network parameters are updated using Adam optimizer with a learning rate of $10^{-3}$. After each time step $k$, the networks are trained with a mini-batch of 256 transitions $(s, a, r, s')$.

**Hyperparameters**. To keep comparisons in this work fair, we set all common hyperparameters (network layers, batch size, learning rate, discount factor, number of agents, etc) to be the same for comparison within the same methods and different methods.

For TD3, target policy smoothing is implemented by adding $\epsilon \sim \mathbb{N}(0, 0.2)$ to the actions chosen by the target actor-network, clipped to $(-0.5, 0.5)$, delayed policy updates consist of only updating the

actor and target critic network every $d$ iterations, with $d = 2$. While a larger $d$ would result in a larger benefit with respect to accumulating errors, for fair comparison, the critics are only trained once per time step, and training the actor for too few iterations would cripple learning. Both target networks are updated with $\tau = 0.005$.

The TD3 used in this study are based on the paper (Fujimoto et al., 2018) and the code from the authors (https://github.com/sfujim/TD3).

| Hyperparameter TD3 | Value |
|---|---|
| Start timesteps | 8000 steps |
| Evaluation frequency | 1e4, 2e4 or 5e4 steps |
| Max timesteps | 1e6, 2e6 or 5e6 steps |
| Exploration noise | $\mathbb{N}(0, 0.1)$ |
| Policy noise | $\mathbb{N}(0, 0.2)$ |
| Noise clip | $\pm 0.5$ |
| Policy update frequency | 2 |
| Batch size | 256 |
| Buffer size | 1e6 |
| $\gamma$ | 0.99 |
| $\tau$ | 0.005 |
| Number of parallel actor | 1 |
| Adam Learning rate | 1e-3 |
| regularization factor | 0.7 |

Table 1: TD3 hyper parameters used for DMC benckmark tasks

The D4PG used in this study is based on paper (Barth-Maron et al., 2018) and the code is modified from TD3. The hyperparameter is from Table 2.

| Hyperparameter D4PG | Value |
|---|---|
| Start timesteps | 8000 steps |
| Evaluation frequency | 1e4, 2e4 or 5e4 steps |
| Max timesteps | 1e6, 2e6 or 5e6 steps |
| Exploration noise | $\mathbb{N}(0, 0.1)$ |
| Noise clip | $\pm 0.5$ |
| Batch size | 256 |
| Buffer size | 1e6 |
| $\gamma$ | 0.99 |
| $\tau$ | 0.005 |
| Number of parallel actor | 1 |
| Adam Learning rate | 1e-3 |
| $V_{max}$ | 100 |
| $V_{min}$ | 0 |
| $l$ | 51 |
| regularization factor | 0.7 |

Table 2: D4PG hyper parameters used for the DMC benckmark tasks

## C FULLSET BENCHMARK RESULTS ON TD3

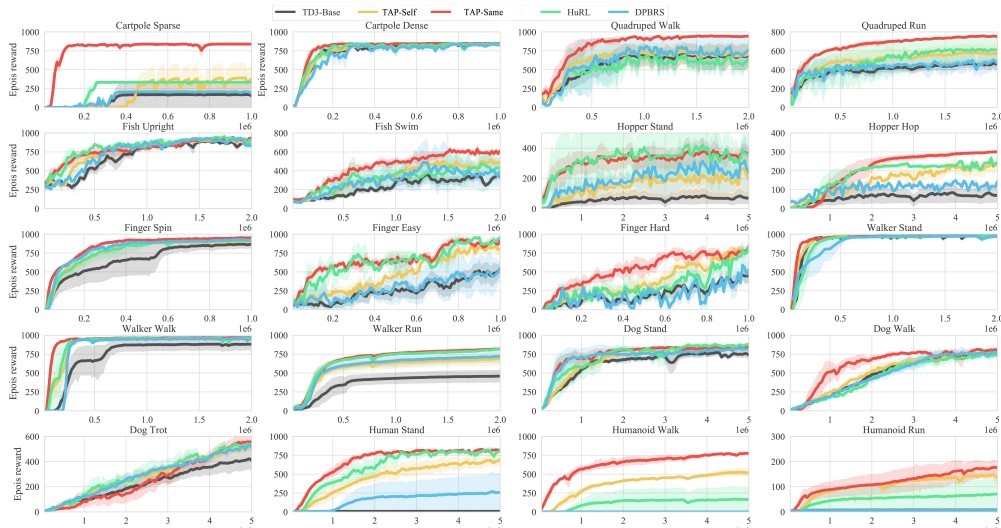

Figure 5: Systematic evaluation of TAP with TD3 base method in DMC environments under **same** task. The shaded regions represent the 95 % confidence range of the evaluations over 10 seeds. The x-axis is the number of steps.

## D D4PG BENCHMARK RESULTS

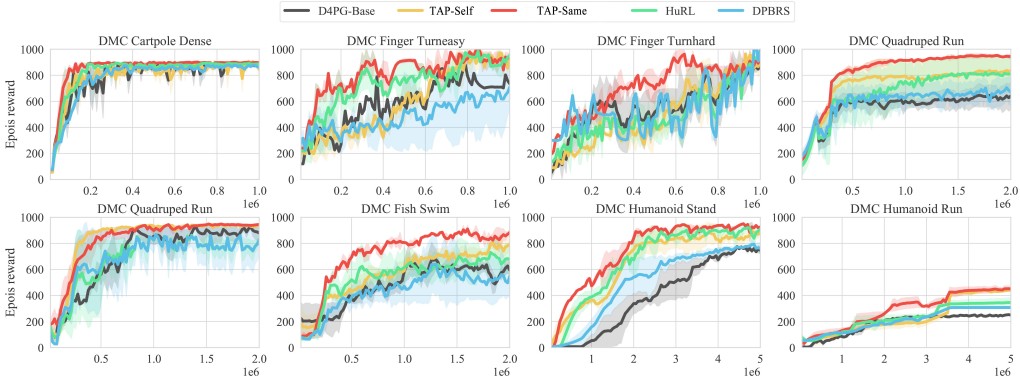

Figure 6: Systematic evaluation of TAP with D4PG base method in DMC environments under **same** task. The shaded regions represent the 95 % confidence range of the evaluations over 10 seeds. The x-axis is the number of steps.

## E THEORETICAL ANALYSIS

**Lemma 1** (TAP Policy Evaluation). Let $r_k$, the stage reward of the target task, be bounded. Given policy $\pi$, let the shaped value function $\mathbb{Q}_i$ be defined as in Equation (9). Then the sequence $\{\mathbb{Q}\}_{i \geq 0}$ of shaped value functions converges to $\mathbb{Q}^\pi$ as $i \to \infty$.

*Proof.* Define the shaping reward $r'_k$ as in Equation (7) and consider the Bellman backup operator as

$$\mathbb{Q}_{i+1}(s_k, a_k) = r'_k + \gamma \mathbb{E}_{s_{k+1} \sim p_\pi, a_{k+1} \sim \pi} \mathbb{Q}_i(s_{k+1}, a_{k+1}). \tag{11}$$

where $s_k \sim p_\pi$ is the state probability induced by the policy $\pi$. Since TAP only directly affects the stage reward of the target task, the standard convergence property of policy evaluation Sutton & Barto (2018a); Haarnoja et al. (2018) leads to the convergence of the sequence $\mathbb{Q}_i$. □

**Lemma 2** (TAP Policy Improvement). Let $\pi_i \in \Pi$ be the policy at iteration $i$. Then the reward shaped value function $\mathbb{Q}^{\pi_i}$ as defined in Equation (9) have that $\mathbb{Q}^{\pi_{i+1}}(s_k, a_k) \geq \mathbb{Q}^{\pi_i}(s_k, a_k)$ for all $(s_k, a_k) \in \mathbf{S} \times \mathbf{A}$.

*Proof.* The result follows from the standard policy improvement theorem in Section 4.2 of Sutton & Barto (2018a). □

**Proposition 1** (Policy Invariance of TAP). Let the prior knowledge be represented in a potential function $\Phi(s)$ as in Equation (5) and denote $\Phi(s) = Q_{\theta'}(s, \pi'(s))$ with respect to policy $\pi$. Let $\mathbb{Q}^*$ and $Q^*$ be the optimal value functions with and without reward shaping, respectively. Then the two optimal policies are the same, namely,

$$\pi^* = argmax_{a \in \mathbf{A}} Q^*(s_k, a) \quad = argmax_{a \in \mathbf{A}} \mathbb{Q}^*(s_k, a) \tag{12}$$

*Proof.* The result is based on Ng et al. (1999), from Lemma 2, the policy improvement property, and by applying Equation (9),

$$\begin{aligned}
\pi^* &= argmax_{a \in \mathbf{A}} \mathbb{Q}^*(s_k, a) \\
&= argmax_{a \in \mathbf{A}}((Q^*(s_k, a) - Q_{\theta'}(s_k, \pi'(s_k)))) \\
&= argmax_{a \in \mathbf{A}}(Q^*(s_k, a))
\end{aligned} \tag{13}$$

Proposition 1 thus holds. □

With the above Lemmas, we have the follow convergence and optimality result of TAP.

**Theorem 1** (Convergence and optimality of TAP). Let $\{\pi_i\}_{i \geq 0} \subseteq \Pi$ be the sequence of policies obtained from repeated application of TAP policy evaluation and TAP policy improvement. Then $\{\pi_i\}_{i \geq 0}$ converges to an optimal policy $\pi^*$. Moreover, for every $(s, a) \in \mathbb{S} \times \mathbb{A}$, and every $\pi \in \Pi$, $\mathbb{Q}^{\pi^*}(s, a) \geq \mathbb{Q}^\pi(s, a)$. Consequently, $\mathbb{Q}^{\pi^*} = \mathbb{Q}^*$, the optimal value function.

*Proof.* Follow the steps in Sutton & Barto (2018b); Haarnoja et al. (2018), let $\pi_i$ be the policy at iteration $i$. By Lemma 2, for any feasible $(s, a)$, the sequence $\{\mathbb{Q}^{\pi_i}(s, a)\}_{i \geq 0}$ is non-decreasing. Also, it is bounded from above by $\mathbb{Q}^{\pi^*}(s, a)$. Hence, it converges pointwise to some limit $\bar{\mathbb{Q}}(s, a) \leq \mathbb{Q}^{\pi^*}(s, a)$. As $\pi_{i+1}$ is chosen from $\pi_{i+1}(s) \in \arg\max_a \mathbb{Q}^{\pi_i}(s, a)$, any limit policy $\pi^*$ of $\{\pi_i\}$ satisfies the inequality $\bar{\mathbb{Q}}(s, \pi^*(s)) \geq \bar{\mathbb{Q}}(s, a)$ for all $a$. That is to say that $\pi^*$ is greedy w.r.t. $\bar{\mathbb{Q}}$, which is the unique fix point of the Bellman optimality operator, i.e., $\bar{\mathbb{Q}} = \mathbb{Q}^*$ and $\pi^*$ is optimal. We thus have $\pi_i \to \pi^*$ and $\mathbb{Q}^{\pi_i} \to \mathbb{Q}^*$. □

Next, we examine how TAP may benefit learning. For that purpose, We consider optimal policy $\pi^*(s)$ with optimal value $Q^{\pi^*}(s, \pi^*(s))$, which denotes the optimal $Q$ value without using reward shaping, and which satisfies the Bellman optimality equation:

$$Q^{\pi^*}(s_k, \pi^*(s_k)) = \mathbb{E}[r_k + \gamma Q^{\pi^*}(s_{k+1}, \pi^*(s_{k+1}))]. \tag{14}$$

**Theorem 2.** Let the stage reward $r_k$ of the target task be bounded by $r_{max}$. Let $\mathbb{Q}(s, a)$ and $Q(s, a)$ (with respective short hand notation $\mathbb{Q}$ and $Q$, and similarly thereafter) as the $Q$-value functions with and without reward shaping, respectively. Assume that the potential function $\Phi = Q_{\theta'}$ containing prior knowledge remains constant, and also that $0 < Q_{\theta'} \leq Q^*$. Set $Q_0 = \mathbb{Q}_0 = 0$. Let $q$ be the $Q$-value that can be reached at step $n_s$ with shaping, and at step $n_{ns}$ without shaping, respectively. Let $\epsilon = \|q - Q^*\|$, we have the following results.

1. $n_{ns} \leq \ln\left(\frac{\epsilon}{\|Q^*\|}\right) / \ln(\gamma)$.

2. $n_s \leq \ln\left(\frac{\epsilon}{\|Q_{\theta'} - Q^*\|}\right) / \ln(\gamma)$.

3. Let $\bar{n}_{ns}$ and $\bar{n}_s$ be the upper bounds of $n_{ns}$ and $n_s$, respectively. Then $\bar{n}_{ns} > \bar{n}_s$.

*Proof.* Consider that it takes $n_{ns}$ updates for $Q$ value to reach $q = Q_{n_{ns}}$ from $Q_0$ without shaping. Under the assumption of prior knowledge $Q_{\theta'}$ with fixed values, and from Equation 9, we can write $q = Q_{n_{ns}} = \mathbb{Q}_{n_s} + Q_{\theta'}$. since it takes $n_s$ updates to reach $q$. Denote the difference between $q$ and the optimal value $Q^*$ as $\epsilon = \|q - Q^*\|$.

Then according to Banach's fixed-point theorem Szepesvári (2022), the convergence of the $Q$ value function without using shaping is:

$$\epsilon \leq \gamma^{n_{ns}} \|Q_0 - Q^*\|. \tag{15}$$

Since $Q_0 = 0$, we can rewrite the inequality as:

$$\frac{\epsilon}{\|0 - Q^*\|} \leq \gamma^{n_{ns}}. \tag{16}$$

Taking the natural logs of both sides,

$$ln\left(\frac{\epsilon}{\|Q^*\|}\right) \leq n_{ns} ln(\gamma). \tag{17}$$

As $0 < \gamma < 1$, $ln(\gamma) < 0$, the inequality becomes

$$n_{ns} \leq ln\left(\frac{\epsilon}{\|Q^*\|}\right) / ln(\gamma). \tag{18}$$

From Theorem 1, after $n_{ns}$ updates, $\epsilon < \|Q_0 - Q^*\|$, from which we have that $ln\left(\frac{\epsilon}{\|Q^*\|}\right) < 0$. Therefore, $0 \leq n_{ns} \leq ln(\frac{\epsilon}{\|Q^*\|})/ln(\gamma)$.

With shaping we have the relationship between $\mathbb{Q}$ and $Q$ as described in Equation 9. Therefore, $\epsilon \leq \gamma^{n_s} \|Q_{\theta'} - Q^*\|$, and $n_s$ obeys the following inequality,

$$n_s \leq ln\left(\frac{\epsilon}{\|Q_{\theta'} - Q^*\|}\right) / ln(\gamma). \tag{19}$$

Because of $0 < Q_{\theta'} \leq Q^*$, we have

$$\|Q^*\| > \|Q_{\theta'} - Q^*\|. \tag{20}$$

As $ln$ is a strictly increasing operator, we have

$$ln\left(\frac{\epsilon}{\|Q^*\|}\right) / ln(\gamma) > ln\left(\frac{\epsilon}{\|Q_{\theta'} - Q^*\|}\right) / ln(\gamma). \tag{21}$$

$\square$

# F    REBUTTAL FIGURE

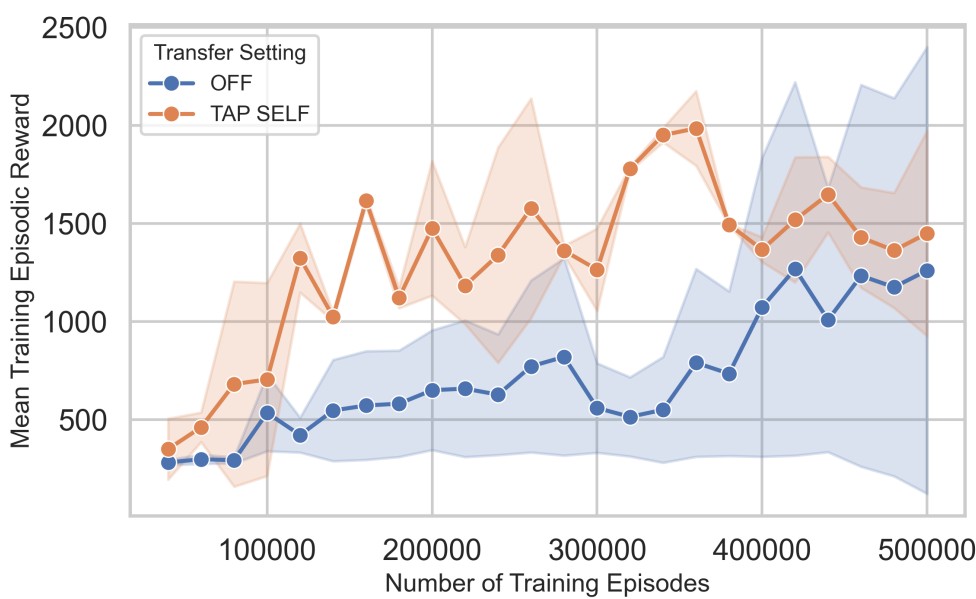

Figure 7: Result of 0.5M training over 2 random seeds on reach-v3 in metaworld benchmark. OFF is the base method of TD3 without shaping. TAP SELF is using current target network as prior to provide shaping reward.

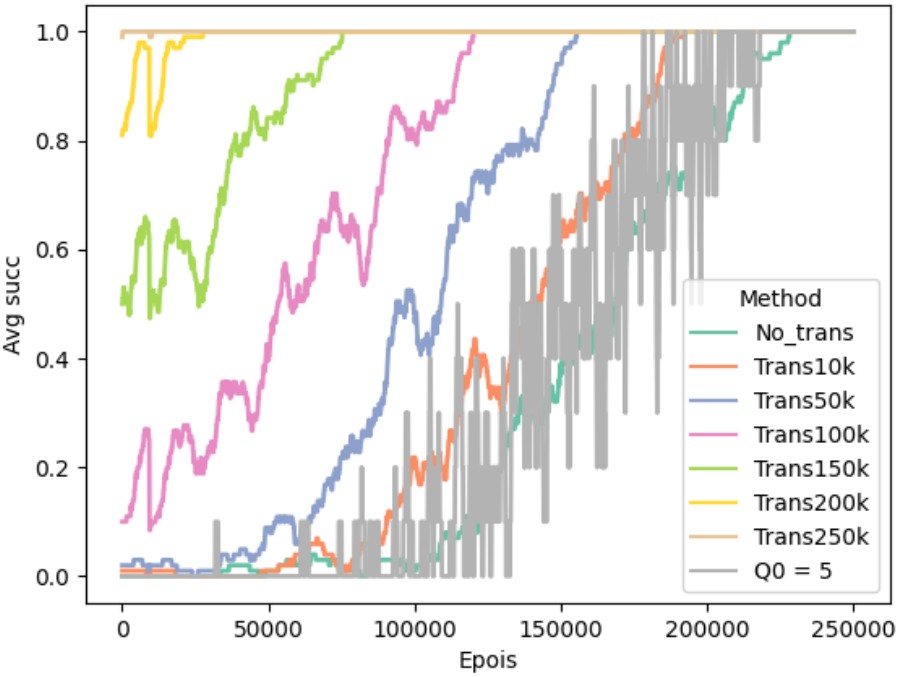

Figure 8: Result of Random initialize Q table to 5

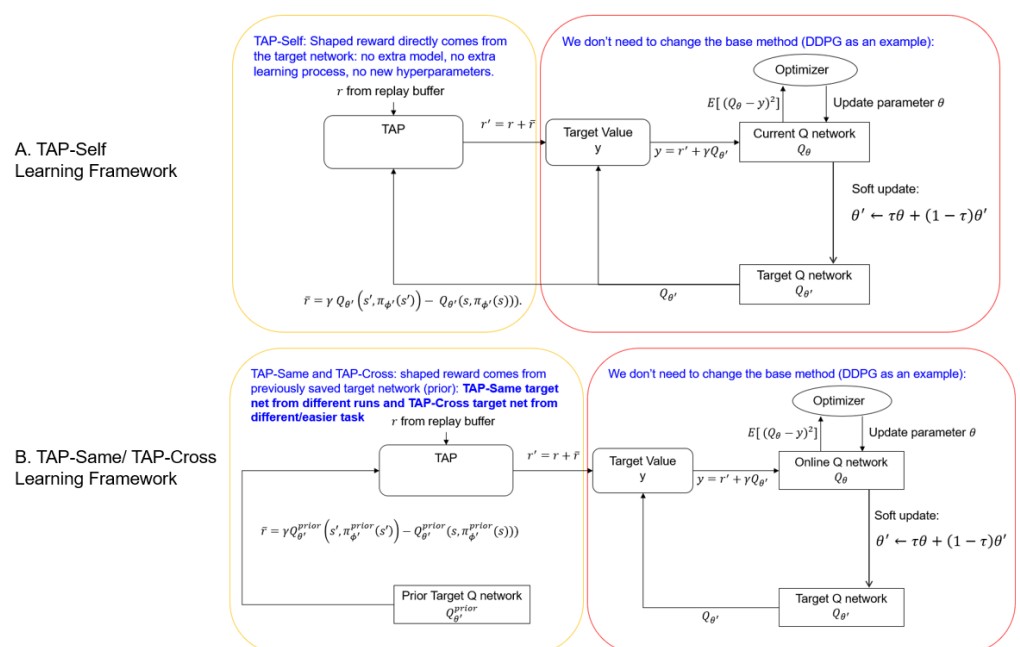

Figure 9: Learning Framework of TAP-Self, TAP-Same, and TAP-Cross. TAP only provides shaped reward signal to formulate the target value $y$, it does not change the learning update process of the base methods.

