# OpenReview forum: "Positive Transfer of Prior Knowledge in Deep Reinforcement Learning via Reward Shaping"
_ICLR.cc/2026/Conference — Submitted to ICLR 2026_

### Official Review · Reviewer_bVbs · 2025-10-16

**Soundness:** 2
**Presentation:** 1
**Contribution:** 1
**Rating:** 2
**Confidence:** 5

**Summary:**

This paper proposed to deal with the knowledge transfer problem through potential based reward shaping with Q function in source domain as the potential. The authors provide some theoretical analysis and experiments on DMC environments.

**Strengths:**

It is a long-standing problem on how to improve the sample efficiency for agents in real-world application and knowledge transfer considered in this manuscript seems to be one potential solution.

**Weaknesses:**

* The idea is not novel. At least this idea is quite similar with [1] and I believe there are more references related to the similar idea included in the manuscripts they refer or refer to them.

[1] Zou, Haosheng, et al. "Learning task-distribution reward shaping with meta-learning." Proceedings of the AAAI Conference on Artificial Intelligence. Vol. 35. No. 12. 2021.

* From the current formulation, this method requires to share the state and action space from the source domain to target domain, which restricts the applicability for more practical scenarios. I'm wondering how to generalize the proposed methods to scenarios without shared states and actions.

* Theoretical analysis are weak and not novel. Policy Invariance can easily be derived from the original reward shaping paper from Ng et al. Convergence rate is of standard textbook style. One tricky part is the authors claimed benefits by assuming we perform Q-iteration from 0 initialization. But in practice this is never the case, and any kinds of positive initialization will be better than 0 initialization. There are no real emphasize on the importance and benefits of knowledge transfer. I would like to see what's the real benefits for the proposed methods.

* For the DMC experiments, there are no clear descriptions about how to generate the cross-task potentials. As not all the DMC environments share the same state action space I'm confused about what's the source task. Also, missing the introduction of cross-task potentials weaken the experimental results.

**Questions:**

To echo the weakness:

* How to compare the proposed methods with the existing work like [1] mentioned in the weakness part?
* What's the real theoretical benefits compared with other randomly initiated non-zero potential function?
* Can you provide a more clear experimental setup for the DMC experiments, like how the potential function from source domain is obtained, and how to deal with different state and action space?

---

> ### Author Response · Authors · 2025-11-22
>
> >The idea is not novel. At least this idea is quite similar with [1] and I believe there are more references related to the similar idea included in the manuscripts they refer or refer to them.
> [1] Zou, Haosheng, et al. "Learning task-distribution reward shaping with meta-learning." Proceedings of the AAAI Conference on Artificial Intelligence. Vol. 35. No. 12. 2021.
>
> We thank the reviewer for reading our paper. We especially thank the reviewer for recognizing that we aim to tackle a "long-standing problem on how to improve the sample efficiency for agents in real-world application and knowledge transfer considered in this manuscript seems to be one potential solution." This is indeed what we intend to achieve, and this work is intended to address this fundamental deep RL problem.
>
> We also thank the reviewer for questioning the novelty of TAP, and how it compares to existing work as well as the new reference [1] provided by the reviewer.
>
> These critical issues actually have given us a chance to highlight the true contribution of our work.
>
> First, we would like to acknowledge that TAP falls within the potential-based reward shaping (PBRS) framework introduced by Ng et al. (1999). We did not introduce any new theoretical concepts within this framework. In this respect, TAP shares the same theoretical foundation as many of the references we cited, as well as the newly added reference [1].
>
> Next, we would like to clearly emphasize that the primary challenge in this area extends beyond what was established by Ng et al. (1999). The key difficulty lies in how to construct or obtain the potential function that defines the shaping signal of the form
> $F\left(s_k, s_{k+1}\right)=\gamma \Phi\left(s_{k+1}\right)-\Phi\left(s_k\right)$, where
> $\Phi$ denotes the potential function. More specifically, the central question is how to determine an appropriate potential function $\Phi$ such that the resulting reward signal can effectively and efficiently guide learning in a new task.
>
> TAP introduces a novel approach for obtaining the prior. Unlike most existing methods, it does not require an additional reward model or a separate learning procedure. Instead, it simply leverages the existing target network.
>
> The method proposed in [1], similar to several existing approaches reviewed in our Introduction, requires both an additional model and a separate learning process. Specifically, it employs meta-learning to construct a distributional prior network based on heuristic data for computing the shaping reward. The overall structure of the method in [1], as described above, is summarized and illustrated in Figure 1 of [1].
>
>
> In our systematic evaluations, we have already included an approach similar to that in [1], namely HuRL, which also relies on an additional model and a separate learning procedure. HuRL uses behavior cloning to learn a heuristic prior from heuristic data for reward shaping. We included HuRL in our evaluation studies because it is more frequently cited than other comparable methods.
>
> >How to compare the proposed methods with the existing work like [1] mentioned in the weakness part?
>
> The two methods, TAP and the one proposed in [1], are not directly comparable. The method in [1] is implemented on DQN, which is primarily designed for discrete tasks, whereas our work focuses on continuous control with continuous state and action spaces. In contrast, the baseline we compared against (HuRL) is specifically designed for continuous control and is therefore directly comparable to TAP.
>
> > From the current formulation, this method requires to share the state and action space from the source domain to target domain, which restricts the applicability for more practical scenarios. I'm wondering how to generalize the proposed methods to scenarios without shared states and actions. how to deal with different state and action space?
>
> Thank you for raising this question. We have indeed considered this and are planning to conduct a systematic study specifically addressing it. However, as this is a new method, our current focus is on establishing a solid foundation by covering applications that share the same state and action spaces. To extend our method to scenarios with different state and action spaces, one could define the potential function within a shared latent space learned through representation alignment (e.g., using contrastive or bisimulation-based encoders) and employ sequence-model or LLM-style adapters to normalize variable-length observations and heterogeneous actions.

---

> ### Author Response · Authors · 2025-11-22
>
> > Theoretical analysis are weak and not novel. Policy Invariance can easily be derived from the original reward shaping paper from Ng et al. Convergence rate is of standard textbook style. One tricky part is the authors claimed benefits by assuming we perform Q-iteration from 0 initialization. But in practice this is never the case, and any kinds of positive initialization will be better than 0 initialization. There are no real emphasize on the importance and benefits of knowledge transfer. I would like to see what's the real benefits for the proposed methods. What's the real theoretical benefits compared with other randomly initiated non-zero potential function?
>
> Thank you for raising this critical question which gives us an opportunity to highlight the significance of the results from our theoretical analyses.
>
> First of all, we would like to emphasize that we never claimed any original contribution regarding the result of policy invariance within the PBRS framework, which was elegantly established by Ng et al. (1999).
> Instead, we would like to respectfully note that nearly all existing theoretical work on reward shaping has primarily focused on demonstrating that reward shaping preserves optimal policies.
>
> While there has been little to no analysis of how shaping signals can be effectively utilized to improve sample efficiency, this gap is precisely what we address in Theorem 2, which is about the importance and benefit of knowledge transfer. This contribution distinguishes our work from existing results. Thus, our paper provides a new theoretical contribution in addition to the innovative idea for determining the potential function.
>
> This leads to our next point: we leverage well-established convergence rate results to provide an important characterization of convergence behavior under both shaped and unshaped conditions. While our analysis builds upon classical convergence rate, the characterization itself is entirely new.
>
> Now specifically about Theorem 2.
> We adopt zero initialization of the Q-values because of the following.
>
> 1) It is standard practice in Q-learning to initialize Q values to 0. In deep RL, although neural networks are typically initialized to small values randomly, the resulting Q-values are generally close to zero with very small standard deviation from 0. As is well known, it is practical to use the last layer bias to achieve 0 value initialization of the Q function. Putting this together, our initialization to 0 is not an issue of concern.
> Consequently, satisfying zero initialization ensures that the condition of $Q_0 < Q^*$  holds uniformly across all state–action pairs as required in Theorem 2.
>
>
> 2) In contrast, random initialization with positive values or simply using any positive initial values cannot guarantee that all such values are accurately specified to be near their optimal values for each and every state-action pairs. That's what learning strives to achieve after all.
> As a result, for some state-action pairs, there is an issue of overly optimistic initialization.
> This is because optimal Q-values vary across state-action pairs: for states far from the goal, optimal Q value is small, and a large random initialization can be overly optimistic.  For example, consider a $Q^*(s,a) = 0.1$, with random initialize $Q_0(s,a) = 1$, it results in optimistic initialization. As is well documented that naıve attempts at optimistic initialization can quickly learn away due to generalization, which can  hurt the performance and  result in large variance
> [Rashid et, al. 2020].
>
> 3)TAP determines the prior from the target network. By Lemma 2, which guarantees the policy improvement property, the prior $Q_{\theta'}$ acts as a monotonically increasing function from 0 upward toward optimal $Q$*  for  all  state-action pairs. This ensures that the assumption of $Q_{\theta'} < Q^*$ remains valid and thus the condition for Theorem 2 holds.
>
> 4)While preparing for the rebuttal, we have obtained a result of initializing Q table to a positive number 5, please refer to Figure 8 in Appendix F. Clearly, initializing the Q values to a positive number not only does not improve performance but actually it introduces large learning variance.
>
> 5)Now we would like to ask the reviewer to revisit our result Figure 1. This empirical result from a simple maze is designed to provide some direct intuition, which helps shed light on Theorem 2 and Remark 2.
>
> Note that, because of the simplicity of the example, the optimal policy can be learned relatively easily. Given a sufficiently long training horizon (e.g., 250k steps), ALL methods (different priors with different quality levels in this case) can reach the optimal policy. However, when the horizon is limited, for example, to 100k steps, the base method without TAP fails to achieve optimal policy. The performance improves by using TAP, the stronger the prior, the more improvement is achieved.

---

> > ### Author Response · Authors · 2025-11-22
> >
> > A similar phenomenon arises in complex deep RL settings. In challenging, high-dimensional tasks such as Humanoid Walk or Humanoid Run, baseline algorithms like TD3 could theoretically solve the task given infinite samples. But in practice, we cannot wait for infinite data. When the training horizon is capped at 1e7 steps, TD3 still fails to learn effectively regardless of that training duration. In contrast, TAP enables successful learning in these tasks using substantially fewer samples. This highlights the theoretical advantage of TAP and also that it allows fine-tuning the residual rather than learning from scratch. This becomes crucial in challenging environments where naive training is infeasible. As shown in Figures 2 and 3, TAP not only accelerates convergence but also makes learning possible in scenarios where standard DRL methods fail.
> >
> > >For the DMC experiments, there are no clear descriptions about how to generate the cross-task potentials. As not all the DMC environments share the same state action space I'm confused about what's the source task. Also, missing the introduction of cross-task potentials weaken the experimental results. Can you provide a more clear experimental setup for the DMC experiments, like how the potential function from source domain is obtained?
> >
> > Thank you for the question. This paper focuses on the use of the target network as a prior or shaping signal within the PBRS framework. For details on the cross-task transfer DMC experiments and results, please refer to the following areas in the paper:
> >
> > (1) Discussions of Figure 3, highlighted by the caption of Figure 3, which states: “Finger Turn Hard environment uses transfer from Finger Turn Easy; Quadruped Run from Quadruped Walk; Humanoid Walk from Humanoid Stand; and Humanoid Run from Humanoid Walk.”
> >
> > (2) The section "Q4. TAP significantly improves performance on challenging tasks by enabling efficient transfer from simpler tasks", lines 405-428.
> >
> > (3) TAP-Cross defined in line 329.
> >
> >
> > Please let us know specifically which aspects are unclear, and we will be happy to clarify them promptly. Thank you.
> >
> > Rashid, T., Peng, B., Boehmer, W. and Whiteson, S., 2020. Optimistic exploration even with a pessimistic initialisation. arXiv preprint arXiv:2002.12174.

---

> ### Comment · Reviewer_bVbs · 2025-11-24
> **Thanks for your response. Additional discussion below.**
>
> I need to admit that I get confused about the meaning of ‘target’ as the authors discussed a lot about knowledge transfer. And I appreciate the illustration within Remark 2 after update. However I still have the questions regarding the differences between using the target value function as typical DQN and using the target value function as described in this manuscript. I want to note that, the manuscript does not provide a detailed algorithm description after shaping the reward in the methodology section. Hence I want to know the following details clearly:
>
> — What’s the supervision signal for critic network after reward shaping? Shall we only change the return and still use the original target critic network to compute TD, or we need to have a different target critic network?
>
> — What’s the difference of the training dynamics between the two method (e.g. how the per-step update is changed in the maze setup)?
>
> — Shall we update the target critic network after some iterations of critic update and reshape everything?
>
> I think this can help the reader understand what the authors really want to do.

---

> ### Author Response · Authors · 2025-11-25
> **Thank you for the detailed questions, which have helped us better highlight the potential of TAP in addressing a fundamental challenge in reinforcement learning.**
>
> >What’s the supervision signal for critic network after reward shaping? Shall we only change the return and still use the original target critic network to compute TD, or we need to have a different target critic network?
>
> >What’s the difference of the training dynamics between the two method (e.g. how the per-step update is changed in the maze setup)?
>
> Please refer to our newly created Figure 9, which provides an intuitive idea how TAP is implemented and how it can be easily integrated into baseline methods. As shown, the only necessary change in the training pipeline is to use $r'$ in the baseline methods.
>
>
> We made a slight modification to what's shown in Figure 9 when transferring knowledge to speed up the maze learning problem. Specifically, since Q-learning does not use target networks, we still leverage priors by adopting the "TAP-Same" approach.  Specifically in this case, the learned $Q$-tables obtained at different stages of training (e.g., at 100k steps or  250k steps, etc.) serve as priors in TAP to compute the shaped reward $\bar{r}$.
> This also shows the flexibility and potentially broad applicability of TAP.
>
> >Shall we update the target critic network after some iterations of critic update and reshape everything?
>
> Hopefully the new Figure 9 helps explain that TAP does not change any learning protocol of the base methods, but only to replace $r$ by $r'$.
>
> With the above said, thank you very much for your prompt review of our rebuttal, we truly appreciate it. Your feedback gave us an opportunity to further clarify the key contributions of our work. We’re pleased that we were able to highlight the important and central role that the concept of the “target” plays in this study, and we are glad that Remark 2 was helpful.
>
> Thank you again for your insightful questions. We look forward to continued productive communication, as we believe this work makes a clear contribution to the field by addressing a "long-standing problem on how to improve the sample efficiency for agents in real-world application and knowledge transfer considered in this manuscript seems to be one potential solution", as you noted in your original feedback.

---

> > ### Comment · Reviewer_bVbs · 2025-11-27
> > **Thanks for the update on the new figure. One question remains.**
> >
> > I’m trying to get more understanding on why the proposed methods work. From Figure 9 it seems we still want to learn the optimal Q function and it just changes the optimization target. Hence I want to know how this optimization target benefits from the original one (e.g. low variance, low initial error etc.) and that’s why I want to know about the difference of training dynamics. Can you provide more elaboration on this?

---

> > > ### Author Response · Authors · 2025-11-30
> > >
> > > First, we kindly direct the reviewer to Remark 1 (Lines 239–248) in the revised manuscript from where we reproduce the following identity
> > >
> > > $ Q^* = Q_{\theta'}$ (a useful prior) + $(Q^* - Q_{\theta'})$(Difference to be learned)
> > >
> > > where $$\mathbb{Q}^*   = Q^{*} - Q_{\theta'}$$ is the shaped Q value.
> > >
> > > As shown, reward shaping by a useful prior has turned the problem of learning the $Q*$
> > > from scratch to the one of learning the difference between $Q*$ and $Q_{\theta'}$ with the proposed shaping. This is an outcome due to using our proposed shaping signal in the optimization target as shown in Equation (9).
> > > Accordingly, since $Q^* - Q_{\theta'} < Q^*$, shaping has allowed the agent to begin learning at a level closer to the optimal solution than learning from scratch.
> > >
> > > As is well known, less accurate critics introduce additional bias into the policy gradient, and noisy critics increase the variance of policy updates. Our proposed shaping method directly reduces the initial Bellman error, yielding a more accurate critic compared to training without shaping. Consequently, this leads to faster, more stable learning and improved sample efficiency.
> > >
> > > Importantly, this reasoning aligns with the theoretical foundation established by Ng et al. (1999), which shows that potential-based reward shaping preserves policy invariance—shaping does not alter the optimal policy that is ultimately learned, only the efficiency with which it is learned.
> > > In our case, incorporating the shaping reward reshapes the training dynamics, resulting in faster, smoother, and more sample-efficient convergence.
> > >
> > > Recognizing that the quality of the prior plays a crucial role in the PBRS framework of Ng et al. (1999), our contribution with TAP is that
> > > 1) TAP leverages an easily obtainable and reliable target that arises naturally during training, and
> > > 2) this prior does not require an additional reward network or a separate reward-learning procedure, unlike most existing approaches.
> > >
> > > In summary, TAP is a novel method accompanied by new theoretical results and insights which are absent from existing reward-shaping techniques. Our empirical evaluations further support these insights and align closely with our theoretical findings.
> > >
> > > Finally, we sincerely thank the reviewer for the thoughtful questions and the prompt engagement with our rebuttals. We greatly appreciate the opportunity to interact and to clarify the novelty and significance of our work. We agree with the reviewer that this research addresses a “long-standing problem on how to improve sample efficiency for agents in real-world applications.” We are confident that our work represents a concrete step toward that goal.

---

### Official Review · Reviewer_4UWa · 2025-10-31

**Soundness:** 2
**Presentation:** 2
**Contribution:** 2
**Rating:** 2
**Confidence:** 3

**Summary:**

The paper proposes Target value As Potential (TAP), a novel reward shaping method.
TAP uses the target critic network (action-value function) from actor-critic RL algorithms as prior knowledge to speed up training.
This is shown both in the same-task and cross-task setting for standard control tasks.
A theoretical analysis/discussion of TAP is provided, giving further insights into the expected benefit of using TAP.

**Strengths:**

- Easy to incorporate into existing methodology, which may facilitate uptake by the community.
- Interesting set of experiments, even if there are not really extensive in terms of number of datasets or baseline methods evaluated.
- Aims for a theoretical analysis to provide further insights in to the method, yet not with any hard guarantees.

**Weaknesses:**

- I did not check the details of the proofs in detail, so can not comment on their correctness. However, I feel they are a somewhat self-fulfilling prophecy, i.e. assuming the target Q-function is a useful prior the number of samples decreases when using it for reward shaping for learning the correct Q-function. Also, this discussion is missing any argument about whether it is necessary to know the Q-function exactly on the full state-action space or just good enough to allow training a good policy training. So overall my complaint would be that the result is somewhat vacuous and does not provide strong arguments for TAP, yet I would look forward to arguments by the authors why this is not the case.
- While it is ok to defer the proof to the appendix, I would greatly appreciate at least a proof sketch in the main paper to get an intuition why the presented result should hold.
- I do not think that Eq.(8) is correct, the expectation over the policy is missing.
- Am I missing something or does one not win anything through TAP in the experiments presented in Figure 1? What incorporating prior knowledge brings is exactly what was needed beforehand to learn the prior. E.g. using a prior obtained from 250k steps I immediately learn to always succeed after very few gradient steps, if I use a prior after 200k steps I need another 50k to close the gap and always succeed. This is not necessarily bad overall, transfering prior knowledge across tasks might still be possible, but it does not seem to bring any benefit when applied on the same task, the overall computation is roughly the same.
- I do not get the reasoning behind the results in Figure 4. (1) are the embedding spaces shared? T-SNE is non-linear and does not allow new datapoints to be projected into the same space. Also, the 3D representation is rather poor to judge, same for the color information. I would rather use 2D maps and work with alpha values to visualize density. Is PRS-IT in the figure TAP? Also, which TAP variant was used for this results? Also, there should be a comparison to the other reward shaping baselines, maybe rather quantitatively then qualitatively in the current visual way.


Presentation and Grammar should be improved. Some examples:
- There is no Remark 1 in the paper, it starts with Remark 2.
- There is no 2) after 1) in line 371
- Line 101: espiecially
- Notation errors such as line 193 where a subscript is used instead of a superscript
- $p_\pi$ in Eq.(3), (4) and (11) never introduced.

**Questions:**

- How exactly is TAP-Self implemented, is it just the current target network?
- Is there any practical purpose of TAP-Same? Don't I just learn the same policy again?
- For the results in Figure 2, what prior run was used for TAP-Same? Did this prior run attain the performance that TAP-Same is attaining or worse? If worse, how can it boost results so strongly?
- Why are only TAP-Same and TAP-cross used in the second set of experiments (Figure 3), I would be interested in the performance of the other baseline or a discussion why they are not applicable.

---

> ### Author Response · Authors · 2025-11-22
>
> >I did not check the details of the proofs in detail, so can not comment on their correctness. However, I feel they are a somewhat self-fulfilling prophecy, i.e. assuming the target Q-function is a useful prior the number of samples decreases when using it for reward shaping for learning the correct Q-function.
>
> We thank the reviewer for reading our paper and for raising thought-provoking questions. Regarding the issue of "self-fulfilling prophecy", we respectfully disagree with the reviewer.
>
> Using target Q-networks has become a standard technique in deep reinforcement learning for stabilizing learning since the successful introduction of DQN in 2013. Nearly all modern off-policy DRL algorithms use target Q net-work(s), which are obtained from delayed copies of the Q network. Therefore, a target Q function also describes the expectation of agent performance as a function of state and action pairs. According to the policy evaluation and policy improvement theorems, the Q-function estimates become more accurate as learning progresses. Con-sequently, the target Q-network accumulates increasingly useful information about the environment’s dynamics and reward distribution over state-action pairs. Therefore, using the target Q-function as a prior is both conceptual-ly sound and practically meaningful.
>
> While our creation of the potential function fundamentally differs from existing approaches, theoretical speaking, existing results have largely stopped at showing that using their shaping signals preserves optimal policies under the PBRS framework, without delving into how sample efficiency may be achieved by using the shaping signal. Our contribution directly addresses this gap. We provide a sample complexity analysis under some reasonable assumptions based on established results of convergence rate (Theorem 2) that quantifies the benefit of TAP-based reward shaping. Please refer to Appendix E for details.
>
> This theoretical characterization of the benefit of TAP is new. It goes beyond intuition and establishes a rigorous justification for reusing learned critics as prior knowledge transferable to new tasks. The idea that a useful prior can accelerate learning has been conceptually supported since Ng et al. (1999), particularly through the lens of policy invariance under reward transformations. However, prior theoretical work has largely stopped at showing that shaping preserves optimal policies, without formally analyzing how a learned Q-function can be reused to improve sample efficiency.
>
> > Also, this discussion is missing any argument about whether it is necessary to know the Q-function exactly on the full state-action space or just good enough to allow training a good policy training.
>
> The Q-function still is over the full state-action space as the shaping signal is only modifying stage reward. We have made this clear in the area under "The TAP method", specifically around Equation (8) and within Theorem 1.   The essence of TAP is that the shaping signal helps us identify "good enough" state-action subspace to improve learning efficiency (please refer to Figure 4).
>
> >So overall my complaint would be that the result is somewhat vacuous and does not provide strong arguments for TAP, yet I would look forward to arguments by the authors why this is not the case.
>
> We sincerely thank the reviewer for keeping an open mind and for being willing to consider our perspective, an attitude we deeply appreciate.
>
> We respectfully disagree with the characterization of our results as vacuous. Our work aims to address a long-standing challenge in deep RL: improving sample efficiency for agents in real-world applications. Our TAP method is based on an innovative idea with concrete backing from modern successes of deep RL using target Q-networks.
>
> Additionally, we support our theoretical claims with a controlled maze experiment (Remark 2, Figure 1), which illustrates two key insights:
> 1) Priors of higher-valued state-action pairs lead to faster convergence and higher cumulative rewards [Ng et,al. 1999, Xiang et,al. 2023]. TAP improves exploration by guiding the agent toward high-value regions in the state-action space. As Figure 1 shows, higher quality priors (transferred Q values at later stages of learning) reduces the likelihood of agents entering the trap at the bottom left corner (0, 3). Alternatively, agents become more likely to visit states with higher values such as (2, 2), (2, 3), (3, 3), (4, 3), and (4, 2). Similarly with Figure 4, TAP has enabled exploration into significantly higher reward-value region.
>
> 2) Our results from both qualitative sample efficiency analysis and strong empirical evaluations offer a concrete justification for TAP and its potential to be an effective transfer RL learning method. It is not merely intuitive, but also easy for integration with existing deep RL pipelines and grounded in formal RL principles.

---

> > ### Author Response · Authors · 2025-11-22
> >
> > > While it is ok to defer the proof to the appendix, I would greatly appreciate at least a proof sketch in the main paper to get an intuition why the presented result should hold.
> >
> > We appreciate the reviewer’s suggestion to include a proof sketch in the main paper. Due to strict page limita-tions, we prioritized presenting additional experimental results that provide valuable empirical insights about and validation of TAP. We would be happy to include the proof sketch once the paper is accepted, as an extra page will then be available.
> >
> > > Am I missing something or does one not win anything through TAP in the experiments presented in Figure 1? What incorporating prior knowledge brings is exactly what was needed beforehand to learn the prior. E.g. using a prior obtained from 250k steps I immediately learn to always succeed after very few gradient steps, if I use a prior after 200k steps I need another 50k to close the gap and always succeed. This is not necessarily bad overall, transfering prior knowledge across tasks might still be possible, but it does not seem to bring any benefit when applied on the same task, the overall computation is roughly the same.
> >
> > Good observation, thank you.  What the reviewer observed resonates what we explained in Remark 1 (originally mislabeled remark 2), which is a perspective on Q-value decomposition and the role of priors in speeding up learn-ing.
> >
> >
> > Now we would like to clarify the purpose of Figure 1. This empirical result based on a simple maze is designed to provide some direct intuition, which helps support Theorem 2 and Remark 2.
> >
> > Note that, because of the simplicity of the example, the optimal policy can be learned relatively easily. Given a sufficiently long training horizon (e.g., 250k steps), ALL methods (different levels of qualities of prior knowledge representation in this case) can reach the optimal policy. However, when the horizon is limited, for example, to 100k steps, the base method without TAP fails to achieve optimal policy. The performance improves by using TAP, the stronger the prior, the more improvement is achieved.
> >
> > A similar phenomenon arises in complex deep RL settings. In challenging, high-dimensional tasks such as Hu-manoid Walk or Humanoid Run, baseline algorithms like TD3 could, in principle, solve the task given infinite sam-ples. But in practice, we cannot wait for infinite data. When the training horizon is capped at 1e7 steps, TD3 still fails to learn effectively regardless of that training duration. In contrast, TAP enables successful learning in these tasks using substantially fewer samples. This highlights the theoretical advantage of TAP and also that it allows learning the residual (which is much easier) rather than learning from scratch. This becomes crucial in challenging environments where naive training is infeasible. As shown in Figures 2 and 3, TAP not only accelerates conver-gence but also makes learning possible in scenarios where standard DRL methods fail.
> >
> > >I do not think that Eq.(8) is correct, the expectation over the policy is missing.
> >
> > Good catch, thank you. We have corrected accordingly. We have also verified other places for consistency.

---

> > > ### Author Response · Authors · 2025-11-22
> > >
> > > > I do not get the reasoning behind the results in Figure 4. (1) are the embedding spaces shared? T-SNE is non-linear and does not allow new datapoints to be projected into the same space. Also, the 3D representation is ra-ther poor to judge, same for the color information. I would rather use 2D maps and work with alpha values to visu-alize density. Is PRS-IT in the figure TAP? Also, which TAP variant was used for this results? Also, there should be a comparison to the other reward shaping baselines, maybe rather quantitatively then qualitatively in the cur-rent visual way.
> > >
> > > First of all, we'd like to correct that "PRS-IT" was intended for "TAP-self". We've made the correction.
> > >
> > > This figure actually has four-dimensional information (s, a, density of (s,a) , reward), and all four are important. The purpose of this figure is to show low-value reward regions are avoided by using TAP, and agents are encour-aged through TAP's guidance to explore high value reward regions.
> > >
> > > Please allow us to first explain how the figure was computed. The figure contains 2M data samples: the first 1M coming from the base method and the second 1M coming from base with TAP. T-SNE map these 2M data sam-ples into same plane.
> > > The reason we need the reward dimension is that it tells us if the agent is exploring in high value regions or other-wise.
> > >
> > >
> > > Similar to [Hong, et al. 2018], we did not compare exploration strategies against other baselines. But a relevant perspective is captured  in Figure 1 and Figure 4 that mainly provide outcomes of effective explanation when we include shaping to improve learning performance and enhances sample efficiency. Actually, any PBRS method that incorporates a beneficial prior should achieve similar exploration advantage as our TAP did. However, the key challenge is how to construct a good prior. In our case, we rely on a reliable target network.
> > > The effectiveness of using our TAP prior has been systematically compared with other methods that obtained their priors differently (Figure 2).
> > >
> > > Finally we would like to mention that we were unable to compare more baseline methods due to hardware limita-tions. Comparing two methods (with and without TAP) by generating the T-SNE visualization requires mapping 2 million data samples onto a single plane. This already has reached  the upper limit of what we can handle; adding additional baselines would exceed memory capacity and prevent seaborn from successfully rendering the plot. Reducing the number of data samples per method is also not ideal, as it would obscure important information about how each agent explores the environment.
> > >
> > > >1. There is no Remark 1 in the paper, it starts with Remark 2.
> > > 2. There is no 2) after 1) in line 371
> > > 3. Line 101: espiecially
> > > Notation errors such as line 193 where a subscript is used instead of a superscript
> > > 4. $p_\pi$ in Eq.(3), (4) and (11) never introduced.
> > >
> > > Thank you, and we have made the corrections accordingly.
> > >
> > > >How exactly is TAP-Self implemented, is it just the current target network?
> > >
> > > Yes. It simply uses the current target network as the potential function for reward shaping.
> > >
> > > >Is there any practical purpose of TAP-Same? Don't I just learn the same policy again?
> > >
> > > Yes, TAP is to learn the same optimal policy. However, in reality, base methods do not always reach the optimal policy as many published benchmark results show. For example, TD3 does not learn Humanoid well. Going back to the maze example in Figure 1, if we limit the training steps to 100k, base method alone cannot reach the opti-mal policy. But with TAP, the optimal policy was reached efficiently.

---

> > > > ### Author Response · Authors · 2025-11-22
> > > >
> > > > > For the results in Figure 2, what prior run was used for TAP-Same? Did this prior run attain the performance that TAP-Same is attaining or worse? If worse, how can it boost results so strongly?
> > > >
> > > > For Figure 2, the prior used for TAP-Same was obtained from a Base method run at 5e6 steps. As shown, this prior performs worse than the final TAP-Same results. The improvement comes from how TAP leverages prior knowledge: it does not simply reuse the prior policy but uses the prior Q-values as a shaping potential, enabling the agent to fine-tune and learn the residual rather than learn from scratch. This mechanism is explained in the derivations of Theorem 2, which shows that shaping reduces sample complexity when training steps are limited.
> > > >
> > > > The same principle can be visualized as illustrated in our maze experiment (Remark 2, Figure 1). When the train-ing horizon is capped at 100k steps, the base method barely learns anything, while transferring that knowledge, TAP (pink line) significantly improves performance. In theory, with infinite samples, both methods can converge to the optimal policy. However, in practice, sample efficiency is critical. TAP significantly accelerates learning under realistic constraints, which is why the boost is so pronounced in complex DRL tasks.
> > > >
> > > >
> > > > Yet from another perspective of exploration, the base method has already explored a great amount of state-action pairs with a good enough value distribution over state-action pairs at the time providing this information as prior. By storing the Q values of respective state-action pairs, and transferring this knowledge to learning the new task, TAP enables the agent to effectively explore high value regions.
> > > >
> > > > >Why are only TAP-Same and TAP-cross used in the second set of experiments (Figure 3), I would be interested in the performance of the other baseline or a discussion why they are not applicable.
> > > >
> > > > We did not compare with the other baselines, HuRL and DPBRS, for the following reasons.
> > > >
> > > >  HuRL derives its heuristic from source-task data through behavior cloning. When evaluting their method, they only performed same task shaping on sparse reacher, humanoid v2, hopper v2, swimmer v2, halfcheetah v2 tasks. In their paper and code, there is no evidence or guidance on  how to transfer from different tasks, for ex-ample, from humanoid stand to humanoid walk.
> > > >
> > > > The DPBRS method depends on episode-level reward signals, which are inherently tied to specific trajectory data. They use episode reward and max episode reward to construct priors, where the episode reward of specific trajec-tories is online information. As such, it is not clear how this information can be transferred across tasks and be helpful to speed up learning. In the reported results, they only performed same task shaping on  Pong and Breakout tasks. Their paper and code did not provide guidance on how to transfer across tasks.
> > > >
> > > > In contrast, TAP does not require external or behavioral data or relying on specific trajectories. Instead, it leverag-es the fundamental learning dynamics (the target networks) that are already available: if some of these dynamics are preserved across tasks, TAP is inherently transferable. Exploring the conditions under which TAP achieves effective transferability—and how this can further benefit algorithms—remains an important direction for future work.
> > > >
> > > >
> > > > Ng, Andrew Y., Daishi Harada, and Stuart Russell. "Policy invariance under reward transformations: Theory and application to reward shaping." Icml. Vol. 99. 1999.
> > > >
> > > > Gao, Xiang, Jennie Si, and He Huang. "Reinforcement learning control with knowledge shaping." IEEE Transac-tions on Neural Networks and Learning Systems 35.3 (2023): 3156-3167.
> > > >
> > > > Hong, Zhang-Wei, et al. "Diversity-driven exploration strategy for deep reinforcement learning." Advances in neural information processing systems 31 (2018).

---

### Official Review · Reviewer_nfHF · 2025-11-03

**Soundness:** 2
**Presentation:** 2
**Contribution:** 2
**Rating:** 6
**Confidence:** 2

**Summary:**

This paper introduces a novel reward shaping method that injects the prior knowledge from other tasks. By simple modification on reward signal using the critic function, the proposed method can effectively transfer the knowledge. Furthermore, as in the main theorem, it is guaranteed that TAP method can converge to the maximum return with less time. In the experiment, the proposed method can effectively adapt to the task more quickly, and also can  transfer the knowledge to the cross-task.

**Strengths:**

1. The overall method is simple and easy to implement. And this paper also proposes the theoretical guarantee on the speed of learning the tasks. Furthermore, the performance of TAP method is quite remarkable compared to the baselines.

**Weaknesses:**

1. What is the main motivation of deriving the TAP method? I know it is quite effective on injecting the prior knowledge, but it is hard to figure out why we should transfer the knowledge in this way. How can we derive this idea without any empirical or theoretical findings?

2. It would be better to show the results on other environments such as Meta-World which consists of multiple skills on robotic manipulation tasks. is the TAP method still effective on robotic manipulation tasks?

**Questions:**

Already mentioned in the Weaknesses section.

---

> ### Author Response · Authors · 2025-11-22
>
> > What is the main motivation of deriving the TAP method? I know it is quite effective on injecting the prior knowledge, but it is hard to figure out why we should transfer the knowledge in this way. How can we derive this idea without any empirical or theoretical findings?
>
> Thank you for reading our paper, and thank you for recognizing that our TAP method "is quite effective".
>
> The motivation for deriving the TAP method is the same as in the classical potential-based reward shaping (PBRS)  framework (Ng et al., 1999): to accelerate learning while preserving policy invariance.
>
> The PBRS augments the original reward with an additional term based on a potential function defined over states, guiding the agent toward desirable regions of the state space. The shaping reward is defined as the difference in potential values described as
> $F\left(s, s^{\prime}\right)=\gamma \Phi\left(s^{\prime}\right)-\Phi(s)$ where $\Phi(\cdot)$ is the potential function.
>
> Although the shaping mechanism is well established, the key challenge in modern RL is constructing a potential function that effectively captures and transfers useful prior knowledge to learning new tasks.
>
>
> As discussed in the Introduction section, most existing efforts on developing priors focus on two types of priors: external-driven and data- or experience-driven. Common in most of the above methods (including reference [1] brought up by Reviewer bVbs), they require learning a separate reward model or a separate potential function model of the environment. As such, they introduce additional complexity, computational overhead, and potential inaccuracies that may hinder the overall learning performance. Additionally, some of those methods are not potential-based, and therefore, they lose the important policy invariance property as in PBRS.
>
> In contrast, our method leverages a key insight: in off-policy reinforcement learning, the target Q-network inherently encodes information about both the environment dynamics and policy performance. Building on this observation, we construct the potential function directly from the target Q-network, enabling intrinsic reward shaping without requiring any additional data or auxiliary models. Moreover, because the target network is updated slowly—a well-established technique for stabilizing learning that has been widely adopted by nearly all off-policy methods since 2013 with the introduction of DQN—it provides a consistent and stable shaped reward over relatively long time horizons.
>
> In terms of theoretical findings, existing results have largely stopped at showing that shaping preserves optimal policies under the PBRS framework, without delving into how sample efficiency may be achieved by using the shaping signal. Our contribution directly addresses this gap. We provide a sample complexity analysis under reasonable conditions based on established results of convergence rate (Theorem 2) that quantifies the benefit of TAP-based reward shaping. It goes beyond intuition and establishes a rigorous justification for reusing learned critics as prior knowledge transferrabl to new tasks. This theoretical characterization of the benefit of TAP is new.
>
> >It would be better to show the results on other environments such as Meta-World which consists of multiple skills on robotic manipulation tasks. is the TAP method still effective on robotic manipulation tasks?
>
> We appreciate the reviewer’s insightful comment and suggestion. Our choice of the DeepMind Control Suite (DMC) was motivated by its status as one of the most widely adopted benchmarks in deep reinforcement learning, including for state-of-the-art (SOTA) methods. Additionally, our evaluations used Humanoid Stand, Walk, and Run, which are generally considered the most difficult tasks to learn in DMC.
>
>
> We agree that extending the evaluation to environments like Meta-World, which focuses on multi-skill robotic manipulation tasks, would provide a more comprehensive assessment of the TAP method’s effectiveness.
> We have thus evaluated the reach v3 task in Meta-World benchmark for training 0.5M steps. The preliminary results that we have obtained are shown in  Figure 7. Clearly, by using our TAP-Self augmented TD3, the learning speed has clearly improved, and the reward trended upward faster with less variance. While the preliminary results have validated our expectations resulting from our theoretical and empirical studies reported in this paper, We plan to perform systematic evaluations using these benchmarks in future research to validate the applicability of TAP in broader and more challenging domains.

---

### Author Response · Authors · 2025-11-22

Dear Reviewers,

Thank you very much for taking the time to read our paper and provide thoughtful feedback. Your comments have given us a valuable opportunity to clarify our contributions. As you read our responses to each of your questions, we kindly invite you to refer to the updated version of our manuscript. We sincerely appreciate your time and consideration, and we would welcome the opportunity to continue this dialogue until all of your questions are fully addressed.

Thank you again for your constructive review and engagement.

Authers

---

### Meta-Review · Area_Chair_tYxC · 2026-01-04

**Summary:**

The paper proposes TAP, a potential-based reward shaping method that uses critic target values as the potential and reports strong empirical speedups on DMC tasks.
Across reviews, the key concerns are concentrated in three areas:
1. Novelty.
Reviewer bVbs argues the core idea is “not novel,” cites close prior work, and views the contribution and presentation as poor.
2. Strength of the theory.
Reviewer 4UWa finds the theory “somewhat vacuous” because it assumes the target Q-function is a useful prior and does not clearly justify when approximate value accuracy suffices for policy learning. Reviewer bVbs similarly describes the theoretical analysis as weak and overly reliant on assumptions (e.g., a particular initialization regime).
3. Experimental clarity and scope:
Reviewers ask for clearer descriptions of how cross-task potentials are constructed (especially when tasks do not share state/action spaces), and for broader evaluation/visualization choices.

Given that two reviewers give a reject score with clear reasons and the remaining reviewer is only marginally positive with low confidence, the reviews support Reject.

**Reviewer Concerns:**

Concerns addressed by the rebuttal:
1. Motivation and derivation for TAP: The authors give a clearer PBRS-based motivation and argues the target Q-network gives a stable, readily available prior.
2. Additional benchmark: The rebuttal adds a preliminary Meta-World Reach-v3 result and claims improved learning speed/variance.
3. Presentation issues: The rebuttal corrects Eq.(8), fixes labeling/typos, and clarifies TAP-Self implementation.
4. Implementation clarity: The rebuttal adds a new framework figure and explains that TAP mainly replaces the target value with a shaped target.
5. Some experimental-detail questions (TAP-Same prior source, why large boosts occur, why some baselines are not used in cross-task experiments): The rebuttal specifies the TAP-Same prior run choice and argues TAP learns a residual rather than reusing a prior policy; it also gives reasons the authors do not run cross-task HuRL/DPBRS comparisons.

Concerns not fully addressed
1. Novelty relative to closest prior work: The rebuttal contests the similarity claim, but the discussion does not fully settle whether TAP is a new algorithmic idea versus a natural instantiation of PBRS using an already-available critic target, and it does not clearly isolate what is fundamentally new compared to cited alternatives (e.g., meta-learned shaping) in a way that would satisfy the strongest novelty critique.
2. Strength of the theory: The rebuttal defends the analysis, but key reviewer objections remain: the guarantees still appear to rely on assumptions that reviewers view as either standard PBRS properties or not tightly connected to practical deep RL training dynamics.
3. Cross-task potential construction when state/action spaces differ: The rebuttal explains certain choices, but the core limitation persists and the reviewer’s confusion about task compatibility and “how to deal with different state and action space” remains only partially resolved.

**Reviewer Scores:**

Reviewer nfHF (6 to likely 6). The rebuttal directly addresses both concerns via PBRS-based motivation and a preliminary Meta-World experiment.

Reviewer 4UWa (2 to likely 3, possibly 4). The rebuttal reduces several concrete objections (correctness/presentation/implementation clarity). However, the reviewer’s core complaint that the theory does not give strong arguments for TAP likely remains.

Reviewer bVbs (2 to likely 2). The rebuttal provides additional implementation and training-dynamics elaboration, but the main novelty and applicability critiques remain.

---

### Decision · Program_Chairs · 2026-01-26

Reject